# Semantic Alignment for Effective Feature Fusion in Real-Time Object Detection

## Abstract

Feature fusion networks are essential components in modern object detectors, aggregating multi-scale features from hierarchical levels to detect objects of varying sizes. However, a significant challenge is that fusing features from different levels often leads to semantic inconsistency due to their distinct representations. While many prior works have attempted to address this, they often incur substantial computational and parameter overhead, limiting their real-time applicability, and in some cases lack generality across different detection architectures. In this work, we propose a novel lightweight semantic alignment module called Feature Interaction NEtwork (FINE). This module refines low-level features by integrating high-level contextual cues via a cross-level attention mechanism prior to fusion. To minimize overhead, FINE combines a kernel-based linear attention with a novel spatial bottleneck design. This design drastically reduces the attention sequence length while preserving the channel-wise semantics essential for effective semantic alignment. FINE is generally applicable to various detectors, including Faster R-CNN, YOLO series, and RT-DETR, and consistently improves detection accuracy without compromising efficiency.

## 1 Introduction

Object detection is a fundamental task in computer vision, which involves simultaneously determining the locations and classes of objects within an image. While early object detectors were primarily based on convolutional neural networks (CNNs) (Ren et al., 2015; Redmon et al., 2016; Tian et al., 2019), more recent approaches have adopted transformer-based architectures to enable end-to-end object detection (Carion et al., 2020; Liu et al., 2022; Zhao et al., 2024). Despite their architectural diversity, robust multi-scale feature representation remains essential for precise detection of objects. To this end, feature fusion networks like Feature Pyramid Networks (FPN) (Lin et al., 2017a) have been widely adopted to combine features from different backbone levels, with numerous improvements proposed to enhance fusion quality and detection accuracy (Liu et al., 2018; Ghiasi et al., 2019; Tan et al., 2020).

Despite the widespread adoption of FPN and its variants, fusion-based approaches often overlook a critical issue: semantic inconsistency between feature levels, as illustrated in Fig. 1(a). In typical feature pyramid architectures, high-level features learn rich semantic contexts while losing spatial information due to repeated downsampling in the backbone network. In contrast, low-level features preserve spatial details but lack sufficient semantic abstraction. Direct fusion of these heterogeneous features via repeated up-sampling and naive addition/concatenation can lead to information conflicts and hinder effective representation learning (Hu et al., 2021). While several refinement methods have been proposed to align features before fusion, as shown in Fig. 1(b–e), these approaches face practical limitations. For example, methods based on deformable operations and channel-wise attention (Huang et al., 2021b; Lu et al., 2022; Dai et al., 2021) often incur high computational costs. In particular, the concatenation of two feature maps (e.g., for learning offsets or channel importance) further increases overhead, which limits their applicability in real-time tasks. Global context aggregation methods (Hu et al., 2021; Huang et al., 2021a) have a complex architecture that is not generally applicable to various detectors. Fusion factor–based methods (Gong et al., 2021; Li, 2024), while effective for detecting small objects, often fail to generalize across diverse detection tasks.

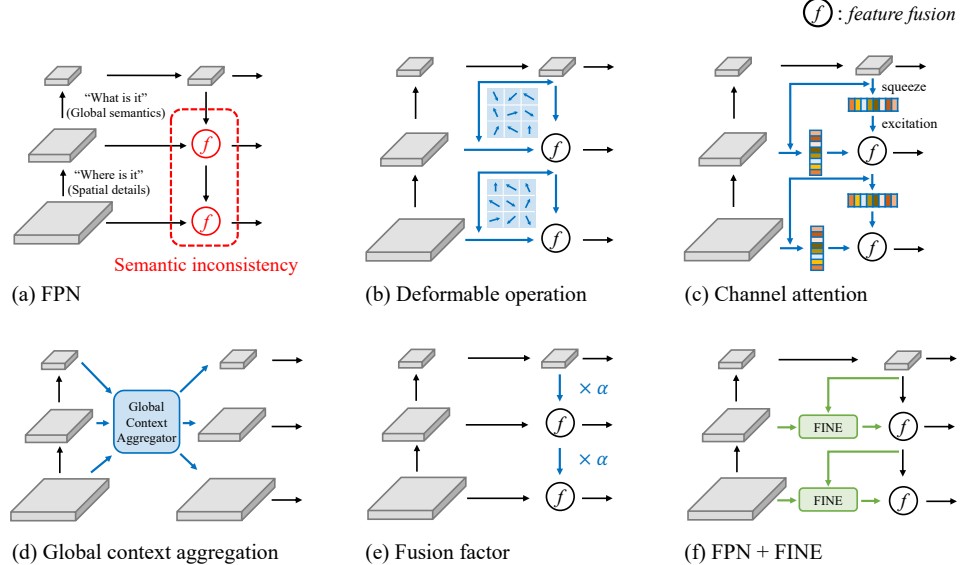

Figure 1: (a) Semantic inconsistency between hierarchical features in FPN. Prior methods addressing semantic inconsistency using (b) deformable operations, (c) channel-wise attention, (d) global context aggregation, and (e) fusion factor. (f) The proposed FINE is a lightweight cross-level attention module that aligns low-level features using adjacent high-level semantics.

Motivated by these limitations, we propose a lightweight semantic alignment module, termed *Feature Interaction NEtwork (FINE)*. As illustrated in Fig. 1(f), FINE operates on low-level features prior to fusion, mitigating semantic inconsistency by allowing high-level features with rich global semantics to guide their refinement via *cross-level* attention. This produces a semantically aligned representation that enables more effective multi-scale feature fusion. In the computer vision domain, attention is generally applied at a single feature level, either as self-attention (Lu et al., 2021; Koohpayegani & Pirsiavash, 2024) or as cross-attention across multiple scale branches within that level (Cai et al., 2023), mainly for classification tasks. In contrast, FINE applies cross-level attention for multi-scale feature fusion in object detection, one of the first approaches aimed at addressing semantic inconsistency in hierarchical features. However, similar to single-level attention, naively applying cross-level attention to high-resolution features leads to quadratic computational and memory complexity. We alleviate this by leveraging kernel-based linear attention (Katharopoulos et al., 2020), reducing complexity to linear. Furthermore, motivated by the observation that semantic information is primarily encoded along the channel dimension (Hu et al., 2018; Woo et al., 2018), we introduce a spatial bottleneck design that drastically reduces the input sequence length while minimizing semantic information loss. This design enables cross-level linear attention to retain the expressiveness of standard softmax attention (Vaswani et al., 2017) while ensuring real-time efficiency.

We demonstrate the effectiveness and generality of the proposed approach for various object detectors, ranging from classic two-stage detectors, such as Faster R-CNN (Ren et al., 2015), to CNN-based one-stage detectors, such as RetinaNet (Lin et al., 2017b), FCOS (Tian et al., 2019), YOLO series (Jocher, 2020; Li et al., 2023; Jocher et al., 2023), and to recent transformer-based real-time detectors, such as RT-DETR (Zhao et al., 2024). Despite the architectural diversity of those detectors, the proposed FINE consistently improves detection accuracy with minimal computational and memory overheads. For instance, on MS COCO dataset, our approach improves mAP of Faster R-CNN by +2.1 with 0.97% increase of FLOPs. Furthermore, when deployed on an edge device, e.g., NVIDIA Jetson Orin Nano, FINE achieves real-time performance, reducing additional FLOPs by 94% and memory usage by 99%, with a negligible 0.1 AP drop compared to standard softmax attention.

The main contributions are summarized as follows.

- We propose **F**eature **I**nteraction **NE**twork (FINE), a general and lightweight semantic alignment module. To the best of our knowledge, this is the first approach to leverage cross-level attention for addressing semantic inconsistency in hierarchical features by guiding low-level features with high-level contextual information prior to multi-scale feature fusion.

- To reduce computational overhead for real-time object detection on edge devices, FINE introduces a kernel-based linear attention paired with a spatial bottleneck design, which significantly lowers spatial computation while preserving channel-wise semantics.

- We show the effectiveness and generality of FINE by integrating it into diverse object detectors, encompassing both conventional (e.g., Faster R-CNN and FCOS) and real-time models (e.g., YOLO series and RT-DETR). On MS COCO dataset, FINE consistently improves detection accuracy with minimal computational overhead.

## 2 RELATED WORKS

### 2.1 FEATURE ALIGNMENT FOR EFFECTIVE FEATURE FUSION

Feature Pyramid Networks (FPN) (Lin et al., 2017a) are widely adopted in object detection for fusing multi-scale features from different backbone layers. Variants such as PANet (Liu et al., 2018), NAS-FPN (Ghiasi et al., 2019), and BiFPN (Tan et al., 2020) enhance this process through bottom-up pathways, neural architecture search, and learnable fusion weights, respectively. However, these methods still fuse high-level and low-level features in a naive manner without explicitly addressing the semantic inconsistency across levels, which can result in suboptimal representation learning.

To address feature misalignment during multi-scale feature fusion, various alignment strategies have been proposed. **Deformable sampling-based methods** (Huang et al., 2021b; Wang & Zhong, 2021; Huang et al., 2021c; Lu et al., 2022) learn pixel-wise offsets to align feature maps of different resolutions and perform deformable operations to produce spatially refined representations. **Channel attention-based approaches** (Dai et al., 2021; Hu et al., 2021) leverage Squeeze-and-Excitation (SE) modules (Hu et al., 2018) to reweight channels based on their importance, facilitating more effective feature integration. **Global context aggregation methods** (Hu et al., 2021; Huang et al., 2021a) capture holistic semantic information across the feature hierarchy and redistribute it to each scale, enhancing semantic consistency across levels. **Other approaches** adopt alternative fusion strategies: AugFPN (Guo et al., 2020) enforces consistent training signals across levels during training, while Soft Nearest Interpolation (SNI) (Li, 2024) uses a fusion factor to control the influence of high-level features during upsampling, aiming to retain texture-rich spatial details from low-level features. Despite their effectiveness, these methods often involve complex designs, introduce considerable parameters and computations, and lack architecture-agnostic compatibility.

### 2.2 LINEAR ATTENTION IN COMPUTER VISION

Linear attention was first studied in the natural language processing domain to reduce quadratic computational and memory complexity with respect to input sequence length (Zaheer et al., 2020; Beltagy et al., 2020; Wang et al., 2020). Among them, kernel-based methods (Choromanski et al., 2020; Katharopoulos et al., 2020) have recently been adopted in high-resolution vision tasks, effectively mitigating the locality bias of CNNs. Most prior work has applied linear attention only to single-scale features (Lu et al., 2021; Koohpayegani & Pirsiavash, 2024) or to multiple scale branches at the same level (Cai et al., 2023) within feature extraction (backbone) networks for image classification. In contrast, we are among the first to integrate linear attention across different hierarchical levels within feature fusion networks for generic object detection. Although Tang & Li (2020) previously explored cross-level attention in the context of salient object detection, their approach was limited to U-Net-based architectures (Ronneberger et al., 2015), which are fundamentally different from FPN variants, and did not consider computational efficiency.

## 3 PRELIMINARIES

In this section, we describe the operational structure of the Feature Pyramid Networks (FPN) (Lin et al., 2017a) and introduce the notations used throughout this paper. As illustrated in Fig. 2(a),

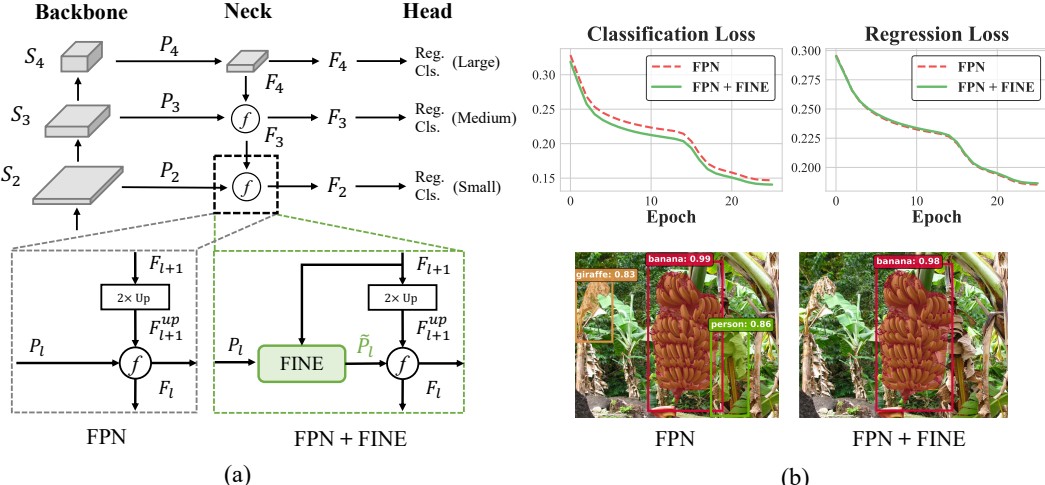

Figure 2: (a) Comparison of feature fusion process with FPN and FPN + FINE. (b) Training loss curves of Mask R-CNN R50. FINE shows better classification optimization through effective category prediction enabled by semantic alignment, while preserving spatial localization despite the spatial bottleneck.

which shows the overall FPN architecture and its multi-scale feature fusion process, the backbone network consists of $L$ stages (four in this example), where feature maps progressively trade spatial detail for richer semantic information. We denote these feature maps as $\{S_1, S_2, S_3, S_4\}$, corresponding to strides of $\{4, 8, 16, 32\}$ pixels relative to the input image. Each feature map $S_l \in \mathbb{R}^{H_l \times W_l \times C_l}$ represents the output of the $l$-th stage, where $H_l$, $W_l$, and $C_l$ denote the height, width, and number of channels, respectively. In particular, $S_1$ provides the highest spatial resolution but is typically excluded from the fusion process due to its high computational cost and limited semantic abstraction. Therefore, FPN and its variants utilize only $\{S_2, S_3, S_4\}$ for multi-scale fusion.

The FPN operates through two key components: a lateral connection and a top-down pathway. In the lateral connection, each $S_l$ is first passed through a $1 \times 1$ convolutional layer to unify the channel dimension to a fixed size $C$, resulting in $\{P_2, P_3, P_4\}$. In the top-down pathway, the higher-level feature $F_{l+1}$ is upsampled using nearest neighbor interpolation in order to match the resolution of $P_l$, yielding $F_{l+1}^{\text{up}}$. This upsampled feature is then fused with $P_l$ via either element-wise addition or concatenation to produce the output $F_l$, which is subsequently used as the high-level feature for the next fusion stage at level $(l-1)$. Through this iterative process, the FPN generates a set of multi-scale fused features $\{F_2, F_3, F_4\}$, each corresponding to a different level of the feature hierarchy. These fused features can be either fed into a bottom-up pathway, as in PANet-style networks (Liu et al., 2018), or directly passed to the detection head to generate final predictions, including bounding box regression and object classification.

## 4 FINE: FEATURE INTERACTION NETWORK

In this section, we present the details of **F**eature **I**nteraction **NE**twork (FINE), as illustrated in Fig. 3. The FINE module is designed to alleviate semantic inconsistency between hierarchical features before fusion. It consists of three key components: (1) a cross-level attention mechanism that explicitly guides low-level features using high-level semantics, (2) kernel-based linear attention to reduce quadratic complexity, and (3) a spatial bottleneck design that drastically reduces the attention sequence length while retaining channel-wise semantic information, enabling real-time deployment.

### 4.1 CROSS-LEVEL LINEAR ATTENTION FOR SEMANTIC ALIGNMENT

To address semantic inconsistency between adjacent feature maps, we propose a cross-level attention mechanism that allows low-level features to attend to high-level ones, capturing informative spatial and channel-wise semantics for alignment. As illustrated in Fig. 2(a), the proposed FINE

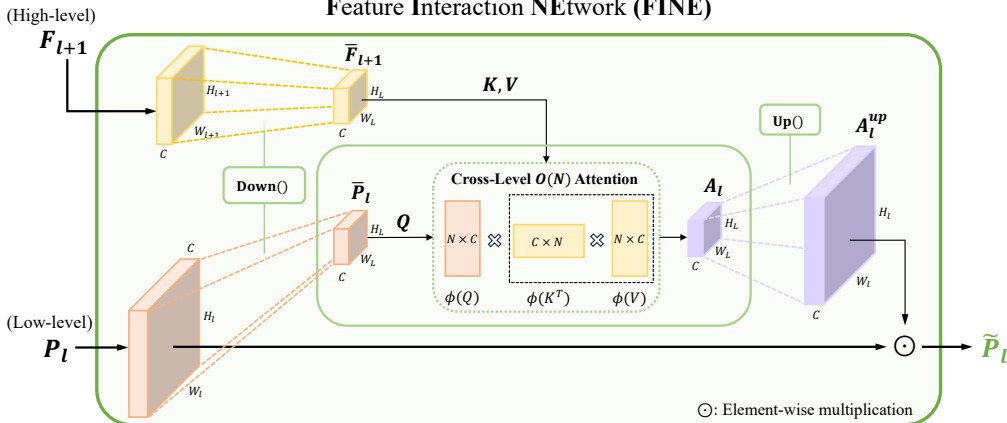

Figure 3: Overview of the **F**eature **I**nteraction **NE**twork (FINE). Given a pair of adjacent feature maps, the low-level feature ($P_l$) attends to the high-level feature ($F_{l+1}$) via a lightweight cross-level attention to mitigate semantic inconsistency. To enhance efficiency and real-time applicability, FINE incorporates a linear attention via kernel approximation, and a spatial bottleneck design tailored for high-resolution efficiency. The output is a semantically aligned feature map ($\tilde{P}_l$), which replaces the original low-level feature during the fusion process.

module uses the high-level feature map as a source of rich global context to refine the low-level representation. Formally, let $P_l \in \mathbb{R}^{H_l \times W_l \times C}$ and $F_{l+1} \in \mathbb{R}^{H_{l+1} \times W_{l+1} \times C}$ denote the low-level and high-level feature maps at level $l$, respectively. The cross-level attention is performed with $P_l$ as the query ($Q$) and $F_{l+1}$ as both the key ($K$) and value ($V$), allowing semantic cues from the high-level feature to guide the low-level feature via contextual weighting across spatial positions and channels. Let $N_l = H_l W_l$ and $N_{l+1} = H_{l+1} W_{l+1}$ be the number of spatial positions in the low- and high-level features, respectively. The cross-level attention is computed as:

$$A_i = \sum_{j=1}^{N_{l+1}} \frac{\text{Sim}(Q_i, K_j)}{\sum_{j=1}^{N_{l+1}} \text{Sim}(Q_i, K_j)} V_j, \tag{1}$$

where $Q = P_l W_Q \in \mathbb{R}^{N_l \times C}$, $K = F_{l+1} W_K \in \mathbb{R}^{N_{l+1} \times C}$, and $V = F_{l+1} W_V \in \mathbb{R}^{N_{l+1} \times C}$. The projection matrices $W_Q, W_K, W_V \in \mathbb{R}^{C \times C}$ transform the input features into query, key, and value embeddings. While multi-head attention is used to improve expressiveness, we describe the single-head case here for clarity. The similarity function $\text{Sim}(\cdot, \cdot)$ measures the relevance between query-key pairs. If defined as $\text{Sim}(Q, K) = \exp\left(\frac{QK^\top}{\sqrt{C}}\right)$, then Eq. (1) becomes the standard softmax attention in Vaswani et al. (2017). The output $A_i$ corresponds to the $i$-th row of the attention matrix $A_l$, which encodes contextually enriched features derived from the high-level feature map. To incorporate this semantic guidance, we perform element-wise multiplication between the original low-level feature and the attention output:

$$\tilde{P}_l = P_l \odot A_l. \tag{2}$$

This operation performs pixel- and channel-wise reweighting on the low-level feature, enabling the attention output to selectively modulate its semantic representation (Hochreiter & Schmidhuber, 1997; Chung et al., 2014). As a result, the semantically aligned feature $\tilde{P}_l \in \mathbb{R}^{H_l \times W_l \times C}$ is obtained and used as the refined low-level input for the feature fusion at level $l$, replacing the original $P_l$.

Similar to standard self-attention, the cross-level attention described above suffers from quadratic complexity with respect to the sequence length of high-resolution low-level features. To address this, we adopt kernel-based linear attention (Qin et al., 2022b), which approximates the softmax similarity using a kernel function $\phi(\cdot)$:

$$\text{Sim}(Q, K) \approx \phi(Q)\phi(K)^\top. \tag{3}$$

We employ an $l_1$-norm layer (Koohpayegani & Pirsiavash, 2024) as $\phi(\cdot)$ to enhance training stability (see Appendix Sec. E for details). With this kernel approximation, the cross-level attention defined

in Eq. (1) can be rewritten as:

$$A_i = \sum_{j=1}^{N_{l+1}} \frac{\phi(Q_i)\phi(K_j)^\top}{\sum_{j=1}^{N_{l+1}} \phi(Q_i)\phi(K_j)^\top} V_j. \tag{4}$$

By exploiting the associative property of matrix multiplication, the cross-level attention can be re-organized as:

$$A_i = \frac{\phi(Q_i)\left(\sum_{j=1}^{N_{l+1}} \phi(K_j)^\top V_j\right)}{\phi(Q_i)\left(\sum_{j=1}^{N_{l+1}} \phi(K_j)^\top\right)}. \tag{5}$$

This reformulation reduces the cross-level attention complexity from $\mathcal{O}(N_l \times N_{l+1})$ to $\mathcal{O}(N_l + N_{l+1})$, yet maintains the representational capability of standard softmax attention, as reported in Table 3.

### 4.2 Spatial Bottleneck Design

Although kernel-based methods reduce cross-level attention complexity to linear, the high spatial resolution of lower-level features still results in long input sequences and substantial memory and computational overhead, making real-time deployment challenging. Motivated by the observation that semantic information is predominantly encoded along the channel dimension (Hu et al., 2018; Woo et al., 2018), we restrict dimensionality reduction to the spatial domain, largely preserves the essential semantics required for alignment. Building on this, we propose a spatial bottleneck design that compresses the spatial resolution of both the low-level feature $P_l \in \mathbb{R}^{C \times H_l \times W_l}$ and the high-level feature $F_{l+1} \in \mathbb{R}^{C \times H_{l+1} \times W_{l+1}}$ to a unified resolution $(H_L, W_L)$, while keeping the channel dimension $C$ intact. Here, $L$ corresponds to the highest level with the smallest spatial size among all backbone stages:

$$\bar{P}_l = \text{Down}(P_l), \quad \bar{F}_{l+1} = \text{Down}(F_{l+1}), \tag{6}$$

where $\text{Down}(\cdot)$ denotes an average pooling operation that downsamples the spatial size to $(H_L, W_L)$. Consequently, since the attention output $A_l$ in Eq. (2) is computed using these down-sampled features, we upsample $A_l$ to restore the spatial resolution of the original low-level feature $P_l$:

$$A_l^{\text{up}} = \text{Up}(A_l; (H_l, W_l)), \tag{7}$$

where $\text{Up}(\cdot)$ denotes a bilinear interpolation to the target size $(H_l, W_l)$. Thus, the semantically aligned feature $\tilde{P}_l$ in Eq. (2) is obtained as follows:

$$\tilde{P}_l = P_l \odot A_l^{\text{up}}. \tag{8}$$

To illustrate, given a $640 \times 640$ input image, the lowest-level feature map at level $l = 2$ contains $N_l = 80^2$ spatial positions, while its corresponding high-level feature map has $N_{l+1} = 40^2$. Although kernel-based linear attention reduces the complexity from $O(6,400 \times 1,600)$ to $O(6,400 + 1,600)$, the cost remains non-trivial for real-time applications. With the spatial bottleneck, both feature maps are downsampled to the highest-level resolution (e.g., $N_L = 20^2$), further reducing the complexity to $O(400 + 400)$, achieving a 90% reduction in overhead. As shown in Table 3, this design not only enables real-time deployment but also preserves the strong representational capacity of standard softmax attention.

## 5 Experiments

### 5.1 Experimental Setup

We conduct experiments on MS COCO dataset (Lin et al., 2014), which consists of 118k training images and 5k validation images. Following standard practice, we evaluate object detection performance using official COCO metrics: mean Average Precision (AP). For fair comparison, both baseline models and ours are evaluated under the same training environments. FPS, latency, and memory usage are measured on NVIDIA Jetson Orin Nano using TensorRT v10.7.0 (NVIDIA, 2024).

Table 1: Performance of the proposed FINE module on various detectors using MS COCO dataset. Models with '*' mark yield slightly lower performance compared to the results reported in the original paper (Zhao et al., 2024), which may stem from differences in experimental setup or unreleased training details. PAN: PANet-style Fusion Network. H-Enc: Hybrid Encoder.

| Model | Fusion Method | #Params | FLOPs | $AP_{50:95}$ | $AP_{50}$ | $AP_s$ | $AP_m$ | $AP_l$ |
|---|---|---|---|---|---|---|---|---|
| Classic Object Detectors | | | | | | | | |
| Faster R-CNN R50 | FPN | 41.8M | 134.4G | 37.0 | 58.5 | 21.1 | 40.3 | 48.2 |
| RetinaNet R50 | FPN | 34.0M | 151.5G | 36.4 | 55.7 | 19.1 | 40.0 | 48.9 |
| FCOS R50 | FPN | 32.3M | 128.2G | 39.2 | 58.2 | 22.1 | 42.4 | 51.3 |
| Faster R-CNN R50 | FPN + *FINE* | 43.1M | 135.7G | 39.1 *(+2.1)* | 60.7 | 23.0 | 43.0 | 49.9 |
| RetinaNet R50 | FPN + *FINE* | 35.7M | 152.9G | 37.3 *(+0.9)* | 56.8 | 20.6 | 40.7 | 49.4 |
| FCOS R50 | FPN + *FINE* | 33.6M | 129.6G | 39.8 *(+0.6)* | 58.7 | 23.3 | 43.1 | 51.2 |
| Real-Time Object Detectors | | | | | | | | |
| YOLOv5-S | PAN | 9.1M | 24.0G | 43.1 | 59.9 | 24.7 | 47.6 | 58.4 |
| YOLOv6-S v3.0 | PAN | 16.5M | 47.2G | 44.3 | 61.2 | 25.0 | 48.7 | 59.7 |
| YOLOv8-S | PAN | 11.2M | 28.6G | 45.0 | 61.8 | 26.0 | 49.9 | 61.0 |
| RT-DETRv1 R18 | H-Enc | 20.2M | 61.7G | 46.5 | 63.8 | 28.4 | 49.8 | 63.0 |
| RT-DETRv1 R50* | H-Enc | 42.9M | 138.0G | 52.8 | 71.0 | 34.2 | 57.3 | 70.0 |
| YOLOv5-S | PAN + *FINE* | 10.2M | 24.9G | 44.2 *(+1.1)* | 61.6 | 24.7 | 49.4 | 59.6 |
| YOLOv6-S v3.0 | PAN + *FINE* | 17.5M | 48.6G | 44.7 *(+0.4)* | 61.2 | 25.2 | 50.0 | 61.8 |
| YOLOv8-S | PAN + *FINE* | 12.3M | 29.6G | 45.8 *(+0.8)* | 62.7 | 27.0 | 51.2 | 62.1 |
| RT-DETRv1 R18 | H-Enc + *FINE* | 21.8M | 63.1G | 47.1 *(+0.6)* | 64.0 | 29.5 | 50.3 | 63.4 |
| RT-DETRv1 R50 | H-Enc + *FINE* | 44.5M | 139.3G | 53.2 *(+0.4)* | 71.6 | 35.9 | 57.4 | 70.2 |

## 5.2 PERFORMANCE ON VARIOUS DETECTORS

To validate the effectiveness and generality of our approach, we integrate the proposed FINE module into various detectors and compare them with the baseline models. The results are shown in Table 1. In Faster R-CNN R50 (Ren et al., 2015), a classic two-stage detector, the proposed FINE module improves the AP from 37.0 to 39.1, with only a 0.97% (+1.3G) increase in FLOPs and a 3.1% (+1.3M) increase in parameters. YOLOv8 (Jocher et al., 2023) is a representative CNN-based one-stage real-time object detector. With the FINE module, the AP of YOLOv8-S increases from 45.0 to 45.8, with only +1.1M additional parameters and +1.0G FLOPs. Unlike previous CNN-based object detectors, RT-DETR R50 (Zhao et al., 2024) is a real-time detector that employs a transformer-based hybrid encoder and decoder for end-to-end detection without NMS post-processing. When our FINE module is applied to the hybrid encoder, the AP improves from 52.8 to 53.2 while incurring additional 0.9% (1.3G) FLOPs and 3.7% (1.6M) parameters. Furthermore, it should be noted that the FINE module incurs only marginal decrease in FPS from 36 to 35.

## 5.3 COMPARISON WITH EXISTING FEATURE ALIGNMENT METHODS

We compare FINE with several representative feature alignment methods on two detector architectures: Faster R-CNN R50 and RT-DETR R18. Fig. 1(b–e) illustrates the key alignment components of each method, while Table 2 presents their detection performance and computational cost. On Faster R-CNN R50, FINE improves AP from 37.0 to 39.1 with a modest overhead (+1.3M parameters, +1.3G FLOPs), demonstrating a favorable trade-off between accuracy and efficiency. While FaPN (Huang et al., 2021b) achieves a slightly higher AP (39.2), it requires more computation (+6.7M parameters, +9.2G FLOPs) mainly due to the introduction of learnable sampling offsets for deformable convolutions, as well as the use of channel attention modules. AdaFPN (Wang & Zhong, 2021) and $A^2$-FPN (Hu et al., 2021) also improve AP (+1.2, +1.3), yet their reliance on adaptive upsampling with learnable offsets and extensive global context modeling leads to substantial computational overheads (+25.0G and +27.4G FLOPs). SNI (Li, 2024) provides a +0.7 AP gain without extra cost via fusion factor control, but is better suited for tasks preserving low-level textures, such as tiny object (Gong et al., 2021) or lane detection (Lv et al., 2023).

Table 2: Comparison with prior feature alignment methods. Methods with '*' are re-implemented and trained either according to the original paper or a PyTorch reference recipe (TorchVision, 2025). 'MGC' denotes the use of only the Multi-level Global Context module from $A^2$-FPN.

(a) Faster R-CNN R50

| Fusion Method | #Params | FLOPs | $\Delta$ AP |
|---|---|---|---|
| *baseline* | 41.8M | 134.4G | - |
| + SNI* | 41.8M | 134.4G | $37.0 \rightarrow 37.7$ |
| + FaPN | 48.5M | 143.6G | $37.9 \rightarrow 39.2$ |
| + AdaFPN | 45.6M | 159.4G | $37.8 \rightarrow 39.0$ |
| + $A^2$-FPN(MGC)* | 44.4M | 161.8G | $37.0 \rightarrow 38.3$ |
| + **FINE** | **43.1M** | **135.7G** | $\mathbf{37.0 \rightarrow 39.1}$ |

(b) RT-DETR R18

| Fusion Method | #Params | FLOPs | $\Delta$ AP |
|---|---|---|---|
| *baseline* | 20.2M | 61.7G | - |
| + SNI* | 20.2M | 61.7G | $38.7 \rightarrow 38.5$ |
| + FaPN* | 21.9M | 71.8G | $38.7 \rightarrow 39.0$ |
| + AdaFPN* | 22.9M | 82.8G | $38.7 \rightarrow 38.3$ |
| + $A^2$-FPN(MGC)* | 21.1M | 68.7G | $38.7 \rightarrow 38.6$ |
| + **FINE** | **21.8M** | **63.1G** | $\mathbf{38.7 \rightarrow 39.3}$ |

Table 3: Comparison of attention variants in the FINE module. This table reports the performance, efficiency, and resource usage of different attention mechanisms integrated into the FINE module, evaluated on NVIDIA Jetson Orin Nano.

| Model | Attention Method | Complexity | #Params | FLOPs | Mem (MB) | FPS | Latency (ms) | AP |
|---|---|---|---|---|---|---|---|---|
| RT-DETR R18 | – | – | 20.2M | 61.74G | 63.03 | 71 | 14.22 | 38.7 |
| **+ FINE** | Quadratic | $O(N_l \times N_{l+1})$ | +1.6M | +22.2G | +280.92 | 36 | 27.81 | 39.4 |
| | Linear | $O(N_l + N_{l+1})$ | | +11.4G | +5.86 | 55 | 18.69 | 39.5 |
| | **Linear + Bottleneck** | $O(N_L)$ | | **+1.3G** | **+3.12** | **68** | **15.07** | **39.3** |

The architectural differences between CNN-based and transformer-based detectors fundamentally affect the efficacy of feature alignment strategies. While CNN-based object detectors preserve spatial dimensions throughout the pipeline and leverage spatial information directly at the detection head, RT-DETR flattens multi-scale features into 1D sequences, concatenates them, and employs cross-attention with learnable object queries. This structural distinction reduces reliance on spatial correspondence and places greater emphasis on channel-wise semantic information. Therefore, methods that primarily target spatial alignment, such as SNI and AdaFPN, fail to generalize effectively in RT-DETR (-0.2, -0.4 AP). In contrast, methods that primarily focus on channel dimension refinement, including FaPN (+0.3 AP) and our FINE module (+0.6 AP), achieve consistent improvements. These findings indicate that channel-wise semantic refinement could be a promising direction for future work, especially in the context of transformer-based detectors.

## 5.4 ANALYSIS OF ATTENTION METHODS IN FINE

To assess the impact of different attention designs within the FINE module, we evaluate three variants in terms of computational complexity, memory usage, and detection performance: (1) standard softmax attention with quadratic complexity, (2) naive linear attention, and (3) linear attention with a spatial bottleneck design. All variants are integrated into RT-DETR R18 and trained under an identical $1\times$ schedule. As shown in Table 3, replacing standard quadratic attention with naive linear attention results in a 49% reduction in FLOPs, a 98% reduction in memory usage (from 281MB to 5.86MB), and a 33% decrease in latency (from 27.8ms to 18.7ms), with similar detection accuracy (39.5 vs. 39.4 AP). To further enhance efficiency, we introduce a spatial bottleneck into the linear attention formulation. This yields additional improvements: FLOPs are reduced by 89%, memory usage by 47%, and latency by 19%, while throughput increases by 24% (from 55 to 68 FPS), all with minimal impact on accuracy (39.3 AP). Importantly, these gains are attained without noticeable loss of information caused by the spatial bottleneck design, which will be discussed in the next subsection 5.5. These results underscore our approach's suitability for real-time, resource-constrained deployment, while demonstrating its effectiveness as a computationally efficient approximation of standard softmax attention.

## 5.5 ANALYSIS OF IMPROVEMENTS IN AVERAGE PRECISION

To better understand the consistent improvements in AP achieved by FINE across different detectors, we analyze the training behavior of Mask R-CNN R50 (He et al., 2017). Fig. 2(b) shows the

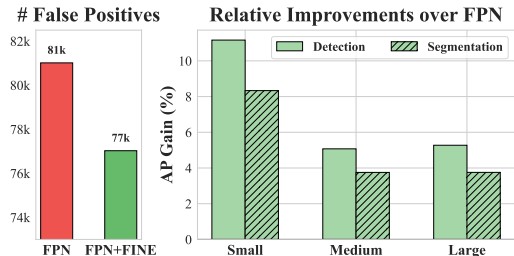

Figure 4: Object detection and instance segmentation results with Mask R-CNN R50. (Left) FINE reduces false positives by 0.8 per image on the COCO 5k validation set (confidence threshold: 0.5). (Right) The AP gain is more pronounced in small objects, likely due to high-level semantic guidance.

Table 4: Performance on VisDrone-DET2019 dataset (input resolution: $800 \times 800$). FINE consistently improves AP on this small-object-dominant dataset.

| Model | Fusion Method | FLOPs | AP | AP$_{50}$ |
|---|---|---|---|---|
| YOLOv5-S | PAN | 37.5G | 28.2 | 45.9 |
| | PAN + *FINE* | 40.8G | 29.0 | 46.9 *(+1.0)* |
| YOLOv6-S v3 | PAN | 72.2G | 27.6 | 44.8 |
| | PAN + *FINE* | 74.5G | 28.1 | 45.6 *(+0.8)* |
| YOLOv8-S | PAN | 44.7G | 28.4 | 46.3 |
| | PAN + *FINE* | 49.2G | 29.1 | 47.2 *(+0.9)* |

training loss curves for the baseline FPN and the FPN + FINE. Notably, the classification loss is consistently lower when FINE is applied, indicating more effective optimization of category prediction. We attribute this improvement to the explicit semantic alignment performed prior to feature fusion, which mitigates semantic inconsistency across hierarchical feature levels. In contrast, the regression loss remains comparable, suggesting that spatial localization is preserved. Although the spatial bottleneck design introduces a downsampling–upsampling pathway, the potential loss of spatial granularity is compensated through the reweighting of original-resolution features via element-wise multiplication, as in Eq. (8)

This enhancement in category prediction leads to a reduction in false positives, as illustrated in Fig. 4(left), evaluated on the MS COCO 2017 validation set (5k images) with a confidence threshold of 0.5. The number of false positives decreases from 81,013 to 77,026, corresponding to an average reduction of 0.8 incorrect detections per image. A considerable portion of these errors can be attributed to ambiguous semantic cues, which may induce high-confidence mispredictions in background areas, as illustrated by examples of false positives in Fig. 2(b). FINE facilitates this reduction by enabling high-level contextual semantics to guide low-level features through cross-level alignment. Additional qualitative analyzes of FINE are provided in Appendix Sec. B.

The improved multi-scale feature fusion resulting from semantic alignment enhances detection across all object sizes, as shown in Fig. 4(right), with the most pronounced gains observed for small objects. To further validate this effect, we evaluate FINE on the VisDrone-DET2019 dataset(Du et al., 2019), which primarily consists of small and densely distributed objects. As reported in Table 4, FINE consistently boosts detection accuracy across YOLOv5, YOLOv6, and YOLOv8 backbones, achieving up to +0.8 AP and +1.0 AP$_{50}$ at an input resolution of $800 \times 800$. Notably, even though these YOLO variants are not specifically designed for small object detection, FINE still delivers consistent improvements in this domain. This indicates that high-level global semantic information effectively guides low-level features, enhancing fine-grained recognition and small object detection. These results further suggest that FINE can be extended to other pixel-wise dense prediction tasks (Appendix Sec. A).

## 6 CONCLUSION

We propose Feature Interaction NEtwork (FINE), a lightweight and general semantic alignment module designed to mitigate the semantic inconsistency that arises in multi-scale feature fusion for object detection. This module refines low-level features by leveraging high-level contextual information via cross-level attention prior to fusion, producing semantically aligned representations that improve multi-scale feature integration. To ensure real-time inference on edge devices, FINE employs kernel-based linear attention and introduces a spatial bottleneck design, which drastically reduces memory and computational costs while preserving channel-wise semantics. Experiments across diverse detector architectures confirm consistent improvements, demonstrating that FINE's lightweight, architecture-agnostic design enables effective multi-scale representation learning.

## REPRODUCIBILITY STATEMENT

We confirm the reproducibility of our results. To facilitate verification during the review process, we provide partial source code and training logs as supplementary material. All materials that are needed to reproduce our results will be released after blind review. We will open source the code as well.

## USE OF LARGE LANGUAGE MODELS (LLMS)

During the preparation of this paper, we used large language models (LLMs) solely for language refinement and grammar correction. LLMs did not contribute to the research ideas, experimental design, or analysis.

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

# Appendix

## A  APPLICABILITY TO OTHER DENSE PREDICTION TASKS

While FINE is primarily designed for object detection, its core principle of enhancing semantic consistency across hierarchical features is also applicable to other dense prediction tasks involving multi-scale feature fusion. To validate this, we integrate FINE into representative frameworks for instance, semantic, and panoptic segmentation using the training recipes provided by OpenMM-Lab (MMDetection Contributors, 2018; MMSegmentation Contributors, 2020). We follow their standard learning rate schedules ($1\times$ for instance and semantic segmentation, and 80K iterations for panoptic segmentation) and input resolutions ($640 \times 640$ for COCO and $512 \times 1024$ for Cityscapes).

As shown in Table 5, FINE consistently improves performance across all three tasks with negligible computational overhead. For instance segmentation on MS COCO (Lin et al., 2014), adding FINE to Mask R-CNN R50 increases box AP by +0.7 and mask AP by +0.5, while Cascade Mask R-CNN R50 (Cai & Vasconcelos, 2019) achieves further gains of +0.8 and +0.6, respectively. In semantic segmentation on Cityscapes (Cordts et al., 2016), incorporating FINE into Semantic FPN (Kirillov et al., 2019a) yields a notable +1.9 mIoU improvement. For panoptic segmentation on COCO, Panoptic FPN (Kirillov et al., 2019a) with FINE achieves +0.7 gains across PQ (Panoptic Quality), SQ (Segmentation Quality), and RQ (Recognition Quality), following the metrics defined in Kirillov et al. (2019b).

These consistent improvements highlight that FINE enhances semantic alignment across hierarchical features prior to fusion, which in turn enables more effective feature fusion. As a result, FINE not only preserves semantic boundaries more accurately but also improves object localization and pixel-level predictions across diverse tasks. This demonstrates its potential as a general-purpose feature fusion strategy for dense prediction.

Table 5: Effectiveness of FINE across dense prediction tasks.

Instance segmentation on MS COCO dataset

| Model | FINE | #Params | FLOPs | Box AP | Mask AP |
|---|---|---|---|---|---|
| Mask R-CNN R50 | X | 44.4M | 144.3G | 38.2 | 34.7 |
| Mask R-CNN R50 | O | 46.0M | 144.9G | 38.9 *(+0.7)* | 35.2 *(+0.5)* |
| Cascade Mask R-CNN R50 | X | 77.3M | 1708.5G | 41.2 | 35.9 |
| Cascade Mask R-CNN R50 | O | 78.9M | 1709.1G | 42.0 *(+0.8)* | 36.5 *(+0.6)* |

Semantic segmentation on Cityscapes dataset

| Model | FINE | #Params | FLOPs | mIoU |
|---|---|---|---|---|
| Semantic FPN | X | 28.5M | 90.9G | 74.5 |
| Semantic FPN | O | 30.1M | 91.7G | 76.4 *(+1.9)* |

Panoptic segmentation on MS COCO dataset

| Model | FINE | #Params | FLOPs | PQ | SQ | RQ |
|---|---|---|---|---|---|---|
| Panoptic FPN | X | 46.0M | 156.7G | 40.2 | 77.8 | 49.3 |
| Panoptic FPN | O | 47.6M | 157.3G | 40.9 *(+0.7)* | 78.5 *(+0.7)* | 50.0 *(+0.7)* |

## B  QUALITATIVE ANALYSIS OF FINE MODULE

To further validate the quantitative reduction in false positives discussed in Fig. 4(left), we present qualitative examples in Fig. 5. These examples demonstrate that FINE reduces false positives such as background textures misclassified as objects or incorrect detections with high confidence. In several cases, FPN produces redundant or spurious detections near object boundaries or background regions, whereas FINE significantly suppresses such errors by aligning low- and high-level features more effectively. These visual results substantiate our hypothesis that the proposed semantic alignment mechanism enhances the reliability of multi-scale feature fusion, thereby leading to more accurate object category predictions.

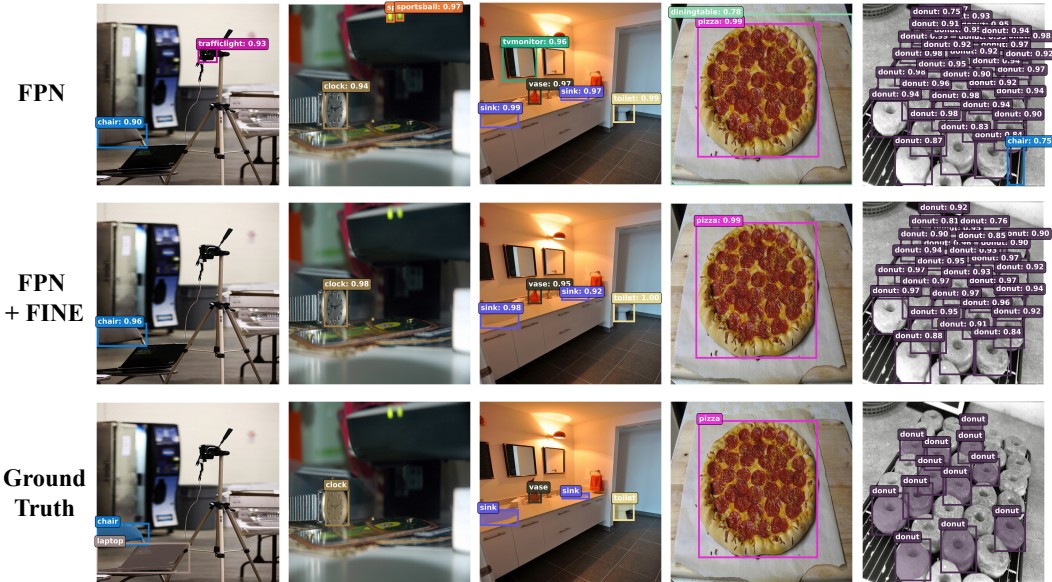

Figure 5: Qualitative comparison of detection results highlighting false positive reduction.

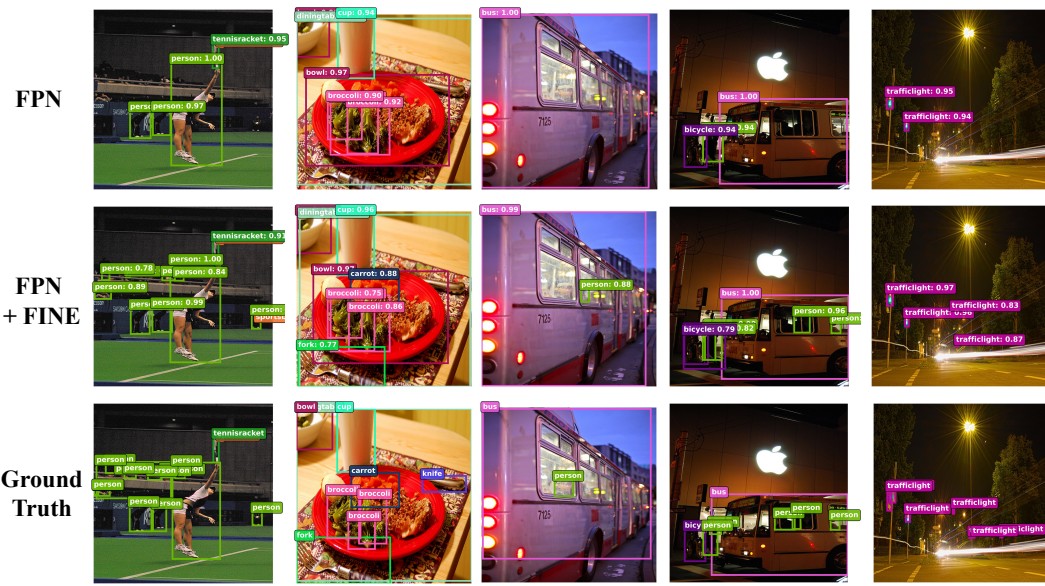

Figure 6: Qualitative comparison of detection results in scenes with complex backgrounds or low lighting.

In addition, Fig. 6 presents examples in challenging visual conditions such as complex layouts and dark lighting. These scenarios often lead to degraded detection performance due to cluttered backgrounds, occlusions, or low visibility. The baseline model frequently misses objects or produces imprecise bounding boxes under such conditions. In contrast, FINE enables more accurate detection by leveraging semantically enriched multi-level features, leading to better localization and classification even in visually adverse scenes.

## C  Unifying Channel Dimensions

As described in Sec. 3, most FPN-based detectors unify the channel dimensions of hierarchical features before fusion, typically by applying $1 \times 1$ convolutions that project each feature map $S_2, S_3, S_4$ into a common embedding space with $C$ channels. This design simplifies subsequent fusion and attention operations, including those in our proposed FINE module, which assumes channel-consistent inputs. Thus, in such architectures, no additional preprocessing is required for channel alignment.

In contrast, YOLO families (Jocher, 2020; Li et al., 2023; Jocher et al., 2023) retain the original backbone output channels $C_2, C_3, C_4$ without enforcing uniformity across scales. To enable cross-level attention among these features, it is necessary to match their channel dimensions, as attention operations require inputs with consistent embedding sizes. Specifically, for each fusion between a high-level feature $F_{l+1} \in \mathbb{R}^{H_{l+1} \times W_{l+1} \times C_{l+1}}$ and a low-level feature $P_l \in \mathbb{R}^{H_l \times W_l \times C_l}$, we apply a $1 \times 1$ convolution to project $F_{l+1}$ into the channel space of $P_l$:

$$\hat{F}_{l+1} = \text{Conv}_{1 \times 1}(F_{l+1}),$$

where $\hat{F}_{l+1} \in \mathbb{R}^{H_{l+1} \times W_{l+1} \times C_l}$ is the channel-adjusted high-level feature used in FINE. By incorporating this lightweight embedding step only when needed, FINE remains broadly applicable across diverse detector architectures while preserving its semantic alignment capability.

## D  Multi-Head Cross-level Attention

We extend the single-head cross-level attention as defined in Eq. (1) to its multi-head variant for improved expressiveness. Let $C$ be the channel dimension and $n_h$ the number of attention heads. Each head operates on a subspace of dimension $d = \frac{C}{n_h}$. Given the low-level and high-level features $P_l \in \mathbb{R}^{H_l \times W_l \times C}$ and $F_{l+1} \in \mathbb{R}^{H_{l+1} \times W_{l+1} \times C}$, we first flatten them into sequences of length $N_l = H_l W_l$ and $N_{l+1} = H_{l+1} W_{l+1}$, respectively. For each head $h \in \{1, \ldots, n_h\}$, we compute the query, key, and value matrices:

$$Q^{(h)} = P_l W_Q^{(h)} \in \mathbb{R}^{N_l \times d},$$
$$K^{(h)} = F_{l+1} W_K^{(h)} \in \mathbb{R}^{N_{l+1} \times d},$$
$$V^{(h)} = F_{l+1} W_V^{(h)} \in \mathbb{R}^{N_{l+1} \times d}$$

where $W_Q^{(h)}, W_K^{(h)}, W_V^{(h)} \in \mathbb{R}^{C \times d}$ are head-specific projection matrices. Each attention head then computes the output as:

$$A_i^{(h)} = \sum_{j=1}^{N_{l+1}} \frac{\text{Sim}(Q_i^{(h)}, K_j^{(h)})}{\sum_{j=1}^{N_{l+1}} \text{Sim}(Q_i^{(h)}, K_j^{(h)})} V_j^{(h)}$$

where $\text{Sim}(Q, K)$ denotes a similarity function. The outputs of all heads are then concatenated and linearly projected to obtain the final output:

$$A_l = \text{Concat}(A^{(1)}, \ldots, A^{(n_h)}) W_O \in \mathbb{R}^{N_l \times C}$$

where $W_O \in \mathbb{R}^{C \times C}$ is the output projection matrix. This multi-head configuration allows the model to capture diverse semantic dependencies across spatial positions and feature subspaces, thereby enhancing semantic alignment while maintaining computational efficiency.

## E  Stabilizing Linear Attention in FINE

We initially implemented cross-level linear attention using a ReLU-based kernel approximation (Choromanski et al., 2020; Qin et al., 2022b). However, training frequently encountered NaNs in both the forward and backward passes, which hindered convergence. To address the instability inherent in kernel-based linear attention, we apply two key stabilization techniques:

1. $\ell_1$-**norm kernel**: Each query and key vector is normalized to have unit $\ell_1$-norm, following the formulation proposed by (Koohpayegani & Pirsiavash, 2024):

$$\phi(Q_i) = \frac{Q_i}{|Q_i|_1 + \epsilon}, \quad \phi(K_j) = \frac{K_j}{|K_j|_1 + \epsilon}.$$

   This kernel balances channel contributions by preventing channels with large magnitudes from dominating the attention output, as discussed in Koohpayegani & Pirsiavash (2024).

2. **RMSNorm-based attention stabilization**: We summarize the key findings of Qin et al. (2022a), who studied the instability of kernel-based attention. The attention matrix $P \in \mathbb{R}^{n \times n}$ is expressed in the unified form

$$p_{ij} = \frac{f(s_{ij})}{\sum_{k=1}^{n} f(s_{ik})}, \quad f : \mathbb{R} \to \mathbb{R},$$

   where the similarity score is $s_{ij} = \phi(Q_i)^\top \phi(K_j)$. The corresponding attention output is given by

$$O = \Delta^{-1} \cdot \phi(Q)(\phi(K)^\top V), \quad \text{with} \quad \Delta = \text{diag}\left(\phi(Q)(\phi(K)^\top 1_n)\right).$$

   Qin et al. (2022a) show that the scaling $\Delta^{-1}$ can cause unbounded training gradients, especially when the similarity scores $s_{ij}$ are small. In particular, the gradient magnitude can grow arbitrarily large:

$$\left| \frac{\partial p_{ij}}{\partial s_{ik}} \right| \leq \frac{1}{4|s_{ik}|}.$$

   To avoid this issue, Qin et al. (2022a) propose removing the normalization by $\Delta$ and applying RMSNorm (Zhang & Sennrich, 2019) to the raw attention output instead:

$$O = \phi(Q)(\phi(K)^\top V),$$

$$O_{\text{norm}} = \text{RMSNorm}(O) = \frac{O}{\sqrt{\frac{1}{d} \sum_{i=1}^{d} O_i^2 + \epsilon}},$$

   where $d$ is the embedding dimension, and $\epsilon$ is a small constant for numerical stability. The key advantage of this replacement lies in achieving *bounded gradients*. Concretely, the following bound holds regardless of input similarity:

$$\left| \frac{\partial \mathcal{L}}{\partial s_{ij}} \right| \leq \frac{3\, c_1\, c_2\, d}{2\sqrt{\epsilon}} < \infty,$$

   where $c_1$ and $c_2$ are bounds on intermediate gradient norms and feature vector magnitudes, respectively. This result ensures gradient stability and enables reliable training.

Together, these modifications eliminate NaN occurrences, stabilize training dynamics, and consistently improve detection performance across different architectures, as summarized in Table 6.

Table 6: Effect of stabilization techniques for linear attention in the FINE module. We compare ReLU-based attention with $\Delta^{-1}$ scaling against $\ell_1$-norm kernel with RMSNorm.

| Model | Fusion Method | FLOPs | Kernel | Rescaling | AP |
|---|---|---|---|---|---|
| Faster R-CNN R50 | FPN | 134.4G | - | - | 37.0 |
| | FPN + FINE | 135.7G | ReLU | $\Delta^{-1}$ | 38.6 |
| | | | $l_1$-**norm** | **RMSNorm** | **39.1** |
| FCOS R50 | FPN | 128.2G | - | - | 39.2 |
| | FPN + FINE | 129.6G | ReLU | $\Delta^{-1}$ | NaN |
| | | | $l_1$-**norm** | **RMSNorm** | **39.8** |
| YOLOv8s | PAN | 28.6G | - | - | 45.0 |
| | PAN + FINE | 29.6G | ReLU | $\Delta^{-1}$ | NaN |
| | | | $l_1$-**norm** | **RMSNorm** | **45.8** |