# OpenReview forum: "Semantic Alignment for Effective Feature Fusion in Real-Time Object Detection"
_ICLR.cc/2026/Conference — ICLR 2026 Conference Withdrawn Submission_

### Official Review · Reviewer_jxd8 · 2025-10-27

**Soundness:** 2
**Presentation:** 3
**Contribution:** 1
**Rating:** 2
**Confidence:** 4

**Summary:**

This paper introduces the Feature Interaction NEtwork (FINE), a lightweight module designed to solve the problem of semantic inconsistency when fusing multi-scale features in object detectors. FINE uses a cross-level attention mechanism to allow high-level, semantically rich features to guide and refine low-level features before they are combined. To ensure suitability for real-time applications, the module incorporates efficient kernel-based linear attention and a novel spatial bottleneck design, which drastically reduces computational overhead while preserving essential semantic information. The authors demonstrate that FINE is a general-purpose solution that consistently improves detection accuracy across a wide range of architectures, including Faster R-CNN, YOLO, and RT-DETR, with minimal impact on performance.

**Strengths:**

1. The paper is very well-written, clearly structured, and easy to follow. The figures and tables are decorative.

2. Main experiments and ablation studies are comprehensive.

**Weaknesses:**

1. The paper's motivation (addressing semantic inconsistency in feature pyramids) is not novel. This is a widely known problem that has been extensively explored by other methods.

2. The proposed FINE module appears to be a straightforward combination of existing techniques, e.g., cross-level attention, kernel-based linear attention, and resizing operations.  The core idea is not new, with precedents like Deformable Attention in DETR models already performing cross-level feature interaction. The work seems more like an engineering assembly of known components than a fundamental contribution.

3. For a paper targeting 2025, the performance gains are not compelling. For example, the improved FINE with RetinaNet-R50 (37.3 AP) is still significantly outperformed by older methods like SEPC [1] with RetinaNet (39.7 AP) from CVPR 2020. The marginal improvements over baselines do not make a strong case for the method's superiority when absolute performance lags.

4. The main results table (Table 1) omits crucial FPS or latency metrics, relying instead on FLOPs, which can be a misleading proxy for actual speed. A comprehensive speed evaluation across all tested detectors is needed to validate the claims of efficiency.

[1] Scale-Equalizing Pyramid Convolution for Object Detection, CVPR 2020.

**Questions:**

I recommend rejecting this paper due to its limited methodological novelty. The proposed FINE module is constructed by combining several well-established techniques, including cross-level attention and kernel-based linear attention, without introducing a new fundamental concept. This assembly feels more like an incremental engineering effort than a significant scientific advance, especially since the core idea of cross-level feature interaction already has precedents in the field. Given that the resulting performance gains are modest, the lack of a core conceptual breakthrough makes the paper's contribution insufficient for a top-tier conference.

---

### Official Review · Reviewer_Ji2y · 2025-10-31

**Soundness:** 4
**Presentation:** 4
**Contribution:** 3
**Rating:** 6
**Confidence:** 3

**Summary:**

The paper proposes FINE, a lightweight cross-level attention module that semantically aligns low-level features using adjacent high-level context before fusion.

To keep the cost suitable for real-time detectors, FINE combines kernel-based linear attention with a spatial bottleneck that compresses spatial resolution while preserving channel semantics. The module is plugged into a variety of detectors (Faster R-CNN, RetinaNet, FCOS, YOLO, RT-DETR) and yields consistent AP gains with small overhead.

**Strengths:**

1. Clear motivation around semantic inconsistency in multi-scale fusion

2. Comparisons against several baselines (FaPN, AdaFPN, A2-FPN, SNI) show competitive or better accuracy-efficiency trade-offs

**Weaknesses:**

1. Significance of Performance Gains: While the improvements are consistent, the magnitude of the gains on several modern, real-time detectors is quite small (e.g., +0.4 AP on RT-DETR R50, +0.4 AP on YOLOv6-S). The most significant gain (+2.1 AP) is on the older Faster R-CNN architecture, which is an old framework for object detection.

2. Novelty of the Core Mechanism: The idea of using attention to align features across different scales or levels is not entirely new. The paper itself cites prior work on cross-level attention. The main novelty seems to lie in the combination of this idea with linear attention and the spatial bottleneck for efficiency in the context of generic object detectors. While this is a solid engineering contribution, the conceptual novelty might be seen as incremental.

3. The proposed method is essentially similar to SAM-DETR (CVPR 2022), which adopts a different method for alignment. The authors should discuss the difference with the proposed method.

**Questions:**

1. Please clarify your novelty against other works.

---

### Official Review · Reviewer_ryDt · 2025-11-01

**Soundness:** 3
**Presentation:** 3
**Contribution:** 3
**Rating:** 6
**Confidence:** 3

**Summary:**

This paper tackles the high computational cost of feature alignment. It argues that prior works are computational expensive for real-time use. The main contribution is a minimal-overhead module (FINE) that combines linear attention with a "spatial bottleneck" design.

**Strengths:**

1. **Strong Efficiency.** The paper validates its low-cost claim. Tables 1 & 2 show minimal FLOPs for significant AP gains, proving its real-time value.

2. **High Generality.** The module works on both CNN (YOLO) and Transformer (RT-DETR) architectures. Table 2 proves it succeeds where older, CNN-focused methods fail, showing its modern relevance.

**Weaknesses:**

**No Ablation for Spatial Bottleneck.** The bottleneck is the core efficiency claim. The paper provides no ablation study on its compression size (e.g., 10x10 vs 20x20). This leaves the key design choice arbitrary and unproven.

**Questions:**

1. Provide the ablation study for the spatial bottleneck's compression size versus AP and FLOPs. This data is required to justify the design.

2. The Table 2 comparison on RT-DETR seems potentially unfair. Competing methods (AdaFPN, SNI) fail - should justify this fail. How can it be a fair comparison if these methods were not optimised?

---

### Note · Authors · 2025-11-12

**Comment:**

Dear Area Chair and Reviewers,

We would like to sincerely thank you for your time and effort in reviewing our manuscript, "Semantic Alignment for Effective Feature Fusion in Real-Time Object Detection" (Paper ID: 6418).

After careful consideration of the insightful feedback, we have decided to withdraw our paper from consideration.

The comments we received are valuable, and we plan to incorporate them to further strengthen the paper for a future submission.

Thank you again for your hard work and constructive suggestions.

Sincerely,

The Authors

**Withdrawal Confirmation:**

I have read and agree with the venue's withdrawal policy on behalf of myself and my co-authors.